# Assessment of the Health Behaviours and Value-Based Health Analysis of People Aged 50+ Who Were Hospitalized Due to Cardiovascular Disease

**DOI:** 10.3390/ijerph18084221

**Published:** 2021-04-16

**Authors:** Izabela Gąska, Katarzyna Sygit, Elżbieta Cipora, Marian Sygit, Anna Pacian, Maryna Surmach, Dorota Kaleta, Adam Rzeźnicki

**Affiliations:** 1Medical Institute, Jan Godek State University in Sanok, 38-500 Sanok, Poland; izagaska@op.pl (I.G.); elacipora@interia.pl (E.C.); 2Faculty of Health Sciences, Calisia University, 62-800 Kalisz, Poland; msygit@onet.pl; 3Department of Public Health, Faculty of Health Sciences, Medical University of Lublin, 20-093 Lublin, Poland; annapacian@umlub.pl; 4Head of the Department of Public Health and Health Services, Grodno State Medical University, 230009 Grodno, Belarus; marina_surmach@mail.ru; 5Department of Hygiene and Epidemiology, Faculty of Health Sciences, Medical University of Lodz, 90-647 Lodz, Poland; dorota.kaleta@umed.lodz.pl; 6Department of Social Medicine, Faculty of Health Sciences, Medical University of Lodz, 90-647 Lodz, Poland; adam.rzeznicki@umed.lodz.pl

**Keywords:** health, pro-health behaviours, anti-health behaviour, prevention, patient, 50+ population, chronic diseases

## Abstract

Introduction: The basic determinant of healthy behaviour—among other human behaviours—is the fact that it consistently affects health. Nowadays, health behaviour studies are considered to be an important method of measuring the health of a population. Objective: To assess the health behaviours and value-based health analysis of people aged 50+ who were hospitalized due to cardiovascular disease, depending on the selected descriptive variables. Materials and methods: The study was conducted between April 2018 and December 2018 among 411 subjects aged 50+ who were hospitalized due to cardiovascular disease at the Independent Public Health Care Unit in Sanok (Podkarpackie voivodship in Poland). The method used in the study was a diagnostic survey. The study used the authors’ survey questionnaire and two standardized tests: Inventory of Health-Related Behaviour (IHB) and List of Health Criteria (LHC). A statistical analysis was carried out in the R program, version 3.5.1. The obtained results were subjected to thorough statistical analysis using the following tests: Student’s *t*, Mann–Whitney U, ANOVA, Kruskal–Wallis, Fisher’s Least Significant Difference (LSD), Pearson, and Spearman. Results: The strongest correlation between health status and health behaviours (according to the IHB questionnaire) was in the area of ‘health practices’, while the lowest correlation was found in the areas of ‘correct eating habits’ and ‘preventive behaviours’. Based on the LHC questionnaire, the most important health criteria according to the subjects were ‘not feeling any physical ailments’; ‘having all body parts functional’; ‘feeling well’; ‘eating properly’; and ‘infrequent need of going to the doctor’. A positive correlation was found in the group of respondents where the ‘preventive health behaviours’ were more intense; herein, the more important criterion for the respondents was ‘eating properly’. Conclusions: Respondents aged 50+ and hospitalized for cardiovascular diseases indicated (based on the IHB questionnaire) that health behaviours in the area of ‘health practices’ had the strongest correlation with their health, while the lowest correlation was found in the areas of ‘correct eating habits’ and ‘preventive behaviours’. According to the respondents, the most important criteria determining health (according to the LHC questionnaire) included ’not feeling any physical ailments’; ‘having all body parts functional’; ‘feeling well’; ‘eating properly’; and ‘infrequent need of going to the doctor’. Based on the information collected from the respondents, it was found that the most important criteria determining health depended on selected descriptive variables, such as age, gender, place of residence, education, and marital status.

## 1. Introduction

Due to the development of the demographic situation in Poland, the issues of reducing morbidity and mortality due to cardiovascular diseases occupy an important place in scientific publications. Cardiovascular diseases have been the greatest threat to the lives of Poles for over 50 years, although the share of these diseases in the total number of deaths has decreased by over 7% over the last 20 years. Undoubtedly, the observed changes have been influenced primarily by the change in health behaviours, especially eating habits and regular preventive examinations [1,2,3].

However, despite the improvement in the epidemiological situation, in the last two decades the mortality rate due to cardiovascular diseases, especially the mortality rate (in people aged below 60), still has been high in Poland. Given the current disease trends and the pace of ageing of the Polish population, it is estimated that the number of deaths due to cardiovascular diseases in 2022 will exceed 200,000 [4,5,6].

The abovementioned epidemiological data justify the need to conduct research on the health behaviours of people of different ages in terms of assessing their health behaviours, and subsequently to take appropriate steps to eliminate incorrect health behaviours that may lead to, among others, cardiovascular diseases.

Nowadays, health behaviour studies are considered to be an important method of measuring the health of a population [1,2,3,4,5,6]. The choice of specific behaviours depends on the importance of health in one’s system of values [1,2,3,4,5,7]. It should be emphasized, however, that the health status is conditioned by many factors, such as behavioural factors, genetic conditions, and health care [4].

Health behaviours are defined as activities aimed at maintaining or regaining health. Health behaviours are a form of a person’s activity aimed at achieving health goals. The aim of the activities undertaken by healthy people is to maintain or increase health resources, while in patients it is to improve their health [2,3,8,9]. These behaviours include behaviours aimed at maintaining well-being; behaviours protecting health; behaviours reducing the risk of losing health; activities of those who noticed disease symptoms in themselves—in order to accurately assess their own health condition and take preventive measures; and activities of a person describing themselves as sick—to improve their state of health and to improve their state of health through a treatment process [8,9,10]. Health behaviours are reactions to all health-related situations, as well as habits and deliberate activities. Each person permanently, intentionally, consciously, and independently makes choices about behaviours that have a positive or negative impact on health. An important determinant of the decisions made is life experience and knowledge in the field of health and disease [1,2,3,4,5,8]. Determinants of choosing health behaviours include age, gender, life goals, social situation (education, social roles, origin, material situation, and place of residence), and cultural situation (world view, family and national traditions, and customs) [1,2,3]. Health behaviours that positively influence human health—apart from one’s self-control, responsibility for one’s own health, and a positive attitude—are mainly a healthy diet, regular physical activity, and an adequate amount of sleep per day. Disease risk factors and thus negative health behaviours include smoking, incorrect diet, low or no physical activity, and alcohol abuse [4,5,11,12,13]. The unsatisfactory health situation of the Polish population, with a tendency to deteriorate in some spheres, requires specific health and preventive measures. The study and analysis of health behaviours create an opportunity to diagnose the areas of ignorance in the field of health care, to determine the risks and harmfulness, as well as to determine the scale of the problem in order to prevent irregularities or pathologies [4,12].

It should be noted that the contemporary society is ageing [10,11,12,13,14]. Life expectancy in Poland increases but it remains significantly shorter than the average in the European Union (EU) by approximately 4 years. The development of civilization and an increasing life expectancy have become the main factors that contribute to the unsatisfactory health condition of the society, especially in the 50+ group [2,15].

The ageing period is characterized by the intensification of physical changes, which are manifested mainly by burdensome ailments and multiple diseases. The biggest threat to life related to incorrect health behaviours are cardiovascular diseases, which are a serious problem in terms of medicine, social life, and economy. Cardiovascular diseases have an epidemiological nature and are therefore referred to as ‘civilization diseases’. They are the most common causes of death in Poland, Europe, and worldwide. They often cause physical and mental disability and require huge financial funding, both in terms of public finances and individual patient spending [16,17,18,19,20,21,22].

Hence, strengthening the correct and changing the incorrect actions that condition one’s health has a significant impact on the increased quality of life, despite the unstoppable changes in the physical functioning of the ageing people [23,24,25,26]. Assessment of the level and the quality of health behaviours allows for undertaking educational activities, care, and treatment towards the senior population [24,25,26]

However, it should be emphasized that the key to maintaining and improving health is prevention.

Incorrect lifestyle is a factor underlying many chronic diseases. A study by Italian researchers has shown that more than half of the Italian population does not meet the WHO thresholds for at least moderate physical activity. The study assessed the impact on health and healthcare expenditure of seven public health policies aimed at promoting exercise and physical activity against a normal business scenario. The assessed policies included promotion of active transport, interventions at sedentary workplaces, investments in sports and recreation, mass-media campaigns, prescription of physical activity in primary care, school-based interventions, and mobile apps. The researchers concluded that public policies promoting physical activity could improve the health of the population and save healthcare expenditure, which would help avoid hundreds of cases of cardiovascular disease and diabetes per year, and dozens of cancer cases [27].

A similar situation in terms of a low level of physical activity can be observed in other countries, e.g., in Poland, Germany, the UK, and Belgium. Hence, appropriate steps should be taken to change the incorrect lifestyle of the society and thus prevent the occurrence of many dangerous chronic diseases in the future [28]

The aim of this study is to assess the health behaviours and value-based health analysis of people aged 50+ who were hospitalized due to cardiovascular disease, depending on the selected descriptive variables. The main hypothesis is as follows: Among the variables of the Inventory of Health-Related Behaviour (IHB), health practices had the strongest correlation with health, while understanding health, according to the List of Health Criteria (LHC), significantly affected the health behaviours of people aged 50+ who were hospitalized for cardiovascular diseases.

## 2. Materials and Methods

### 2.1. Study Design

This publication is another analysis of the results of studies conducted within a research project carried out on the territory of Poland and concerning the quality of life and health behaviours of people aged over 50 years old who were hospitalized due to cardiovascular diseases in Poland.

This study was triggered by the deteriorating health situation of people aged 50+ in Poland in the last two decades. The highest percentage of morbidity and mortality is caused by cardiovascular diseases. The risk factors for cardiovascular diseases include, among others, age, gender, genetic predisposition, as well as incorrect health behaviour, which became a starting point of this study. Standardized questionnaires—the Inventory of Health-Related Behaviour (IHB) and List of Health Criteria (LHC)—were used in the study; they are designed to study the health situation of people of different ages. These tools offer a detailed picture of health behaviours and an analysis of the respondents’ health.

### 2.2. Studied Population

The study was conducted between April 2018 and December 2018 among 411 subjects aged 50+ who were hospitalized due to cardiovascular disease at the Independent Public Health Care Unit in Sanok (Podkarpackie voivodship in Poland).

The simple random selection method was used, and the study was carried out using 3 survey questionnaires, while the remaining data were obtained from the so-called patients’ ‘health cards’. The research tools allowed the authors to analyse the health situation of the respondents in the last 10 years.

Based on those records, the authors verified the existing cardiovascular diseases and obtained the necessary data for the calculation of indicators such as BMI. The (anonymous) questionnaires, along with patients’ informed consent to participate in the study, were completed by the patients in accordance with the study instructions presented to them earlier.

The criteria for the study group were the following: age—50+; and gender—women and men diagnosed with cardiovascular disease and who were hospitalized due to the diseases in the Independent Public Health Care Unit in Sanok, Podkarpackie voivodship, Poland. The criteria for exclusion from the study group were age under 50, no diagnosed cardiovascular diseases, and gender—not hospitalized men and women.

#### 2.2.1. Description of Research Tools

The method used in the study was a diagnostic survey. The study employed 3 research tools:Authors’ 3-part survey questionnaire. The first part focused on social information; the second part—health of the subjects; and the third part—their lifestyle.The second tool was the standardized questionnaire ‘List of Health Criteria’ (LHC) by Zygmunt Juczyński. LHC contained 24 statements describing the positive elements of three dimensions of health: physical, mental, and social; its results help determine respondents’ preferences in terms of health determinants, discover what the respondents understand by the concept of health, what it means for a given individual to ‘be healthy’, and discover the extent to which health is equated to state, process, or property. The respondents reported their preferences, indicating which of the given statements were important in their health assessment, and which among the selected statements were the most crucial. It resulted in a ranking of all 24 health criteria which characterized the study group. To interpret the results, one needed to consider the distribution of the number of ranks of the individual health criteria. LHC may be useful in activities aimed at modifying health behaviours, as well as in therapy and rehabilitation.The third research tool was the standardized ‘Inventory of Health-Related Behaviour’ (IHB) developed by Zygmunt Juczyński. It consists of 25 statements describing various types of health-related behaviours. It enabled an overall assessment of the intensity of healthy behaviours, as well as 4 categories of health behaviours, namely, correct eating habits, preventive behaviour, positive mental attitude, and health practices. The value of the general indicator of health behaviours ranged from 24 to 120 points. The higher the indicator, the greater the intensity of the declared healthy behaviours. After conversion into standardized units, the indicator was subjected to interpretation as sten scores. Scores 1–4 were considered low, 5–6 average, and 7–10 high. For this scale, normative values for various age and social groups as well as for the healthy and the ill were determined. The author of the questionnaire provided sufficient reliability for the overall score of the IHB (α = 0.85) and for the individual scales (Cronbach’s alpha index ranged from 0.60 to 0.65) [29].

#### 2.2.2. Consent of the Bioethical Committee to Conduct the Study

The study was approved by the Bioethics Committee of the Medical University of Lodz, under number RNN/156/18/KE.

#### 2.2.3. Statistical Analysis

A statistical analysis was carried out in the R program, version 3.5.1. (The R Consortium, Vienna, Austria).

The analysis of the quantitative variables (i.e., expressed in numbers) was performed by calculating the mean, standard deviation, median, quartiles, minimum, and maximum values. The analysis of the qualitative variables (i.e., not expressed in numbers) was conducted by calculating the number and percentage of occurrences of each value. ‘0′ means zero points, i.e., a situation in which a given criteria was not selected by the respondent.

The comparison of the values of quantitative variables in two groups was made using Student’s *t*-test (when the variable had a normal distribution in these groups) or the Mann–Whitney U test (for a non-normal distribution).

A comparison of the values of the quantitative variables in three or more groups was made using ANOVA analysis (when the variable had a normal distribution in these groups) or the Kruskal–Wallis test (for a non-normal distribution). After detecting statistically significant differences, post-hoc analysis was carried out with Fisher’s Least Significant Difference (LSD) test (normal distribution) or Dunn’s test (non-normal distribution) to identify the statistically significant differences between groups.

Correlations between the quantitative variables were analysed using the Pearson correlation coefficient (when both variables had a normal distribution) or Spearman correlation coefficient (otherwise). The strength of the correlation was interpreted according to the following scheme:

|r| ≥ 0.9—very strong correlation

0.7 ≤ |r| < 0.9—strong correlation

0.5 ≤ |r| < 0.7—medium correlation

0.3 ≤ |r| < 0.5—weak correlation

|r| < 0.3—very week correlation (negligible).

Interpretation based on Hinkle D.E., Wiersma W., Jurs S.G. Applied Statistics for the Behavioural Sciences. 5th ed. Boston: Houghton Mifflin, 2003 [30].

The normality of the variable distribution was tested using the Shapiro–Wilk test. The analysis adopted the significance level of 0.05 (thus, all *p* values below 0.05 were interpreted as significant correlations) [31].

## 3. Results

The characteristics of the study group are presented in Table 1. The group was dominated by men—there were 223 men (54.26%) and 186 women (45.26%). The age structure of the study group: the average age of the subjects was 69.2 (SD = 9.45 and ranged from 50 to 93); the median was 69.

Body weight of the subject group: 157 subjects (38.20%) were overweight and 125 (30.41%) suffered from class 1 obesity.

Place of residence of the study group: the most numerous group was the inhabitants of cities below 100,000—223 people (54.26%)—while there were 183 people in rural areas (44.53%).

Education of study group: the proportion of people with vocational education was the highest—191 people (46.47%)—while 145 people (35.28%) reported secondary education.

In the study group, the majority, i.e., 313 people (76.16%), were professionally inactive, while 84 people (20.44%) were still active.

The structure of marital status: the largest group were married people—311 subjects (75.67%)—followed by widows and widowers—73 individuals (17.76%).

Analysing the incidence of cardiovascular diseases amongst subjects, it was found that 262 people (63.75%) had coronary heart disease, which was followed by atherosclerosis, 240 people (58.39%), and hypertension, 162 people (39.42%) (Table 1).

Analysis of the health behaviour results according to the Inventory of Health-Related Behaviour (IHB) and the analysis of health assessment criteria according to the List of Health Criteria (LHC) showed that people aged 50+ who were hospitalized due to cardiovascular diseases indicated the following as the important health criteria:not feeling any physical ailments;having all body parts functional;feeling well;eating properly;infrequent need of going to the doctor (Table 2).

As a result of the comparison of the most important statements regarding health, taking into account the age of the respondents, it was found that age significantly and positively correlated with the following criteria (*p* < 0.05):drinking little or no alcohol;not getting sick, at most with flu, indigestion (rarely);having healthy eyes, hair, and skin.

It should be noted that the abovementioned health criteria were more important for the elderly respondents.

On the other hand, age correlated significantly and negatively with the following criterion (*p* < 0.05):ability to work without tension and stress.

For elderly patients, this criterion was less important than for the rest of the study group (Table 3).

After analysing the most important statements in the respondents’ opinion, it was found that the perception of 9 out of 24 health criteria significantly depended on gender (*p* < 0.05).

The studied group of women perceived the following health criteria as more important than men:feeling happy most of the time;eating properly;drinking little or no alcohol;not smoking tobacco;ability to work without tension and stress.

Less important criteria for the studied group of women were:
not feeling any physical ailments;not getting sick, at most with flu, indigestion (rarely);having all body parts functional;infrequent need of going to the doctor (Table 4).

After analysing the study results, it was found that the perception of 3 out of 24 health criteria significantly depended on BMI (*p* < 0.05). A post-hoc analysis was performed, which helped discover significant relationships. The analysis showed that:
obese people perceived the criterion ‘eating properly’ as less important than other respondents and obese people perceived the criterion ‘having all body parts functional’ as more important than those with underweight and regular body weight. (Table 5).

After analysing the study results, it was found that the perception of 9 out of 24 health criteria significantly depended on place of residence (*p* < 0.05).

The respondents who lived in cities perceived the following criteria as more important than those living in a rural environment:eating properly and adequate amount of rest, sleep.

On the other hand, the respondents who lived in cities perceived the following criteria as less important: drinking little or no alcohol and not getting sick, at most with flu, indigestion (rarely) (Table 6).

The statistical analysis showed that the perception of 16 out of 24 health criteria significantly depended on education (*p* < 0.05). A post-hoc analysis was performed, which helped discover that people with vocational and secondary education perceived the criterion ‘getting along well with other people’ as less important than people with primary and higher education and people with higher education perceived the criteria of ‘knowing how to solve one’s problems’, ‘ability to work without tension and stress’, ‘ability to adapt to life changes’, ‘ability to control one’s own feelings and desires’, ‘accepting oneself, knowing one’s possibilities and limitations’, and ‘having a job, various interests’ as more important than other respondents. (Table 7).

After analysing the study results, it was found that the perception of 3 out of 24 health criteria significantly depended on the professional activity (*p* < 0.05).

The respondents who were professionally active perceived the following criterion as more important than inactive ones: ability to work without tension and stress.

Meanwhile, the criteria perceived by the professionally active respondents as less important were:getting along well with other people;not getting sick, at most with flu, indigestion (rarely) (Table 8).

The analysis of the comparison of health statements with the marital status of the respondents showed that the perception of 12 out of 24 health criteria significantly depended on marital status (*p* < 0.05).

Married respondents perceived the following criteria as more important than other respondents: eating properly; adequate amount of rest, sleep; and not smoking tobacco. (Table 9).

The analysis of the study results based on the IHB questionnaire showed that the most common health behaviours were related to ‘health practices’, slightly less common in the area of ‘positive mental attitude’, and the least common in the area of ‘proper eating habits’ and ‘preventive behaviours’ (Table 10).

By correlating the level of health behaviours according to the IHB with individual categories from the LHC, a total of 59 statistically significant relationships were observed (*p* < 0.05). A list of these dependencies, sorted from the strongest to the weakest, is presented in the table below (Table 11).

## 4. Discussion and Limitations

The results of the conducted study painted the picture of the health behaviours and health analysis of people aged 50+ who were hospitalized for cardiovascular diseases, depending on the selected descriptive variables.

The study (according to the LHC) showed that the following factors were of great importance in the selection of appropriate criteria determining the health of the respondents: gender, age, education, place of residence, and marital status.

The conducted study has shown that, for the older respondents, the most important factor determining their health is the avoidance of stimulants, while the examined group of women considered the following health criteria as more important than the studied group of men: feeling of happiness and proper nutrition. The respondents from the cities indicated that important criteria for health were proper nutrition, sleep, and rest—more often than their rural counterparts. The education of the respondents had a significant impact on the results obtained: people with higher education chose solving difficult life situations and working without stress as the most important criteria of health. The marital status of the respondents also turned out to be extremely important in choosing the criteria determining health: married respondents considered proper nutrition, as well as sleep and rest as the most important criteria of health.

The study (based on the IHB) has shown that the respondents rated the highest health practices related to proper daily health habits, such as sleep, rest, and regular meals.

Based on the obtained results, the authors initiated a discussion with other authors conducting studies with the same research tools, which helped make a precise summary.

‘Having all body parts functional’ was the most important criterion for people aged 50+ in the study conducted by Nowicki and Ślusarska [8]. It was followed by ‘reaching very old age’. The authors’ own study showed that health was understood by the respondents as *property* and *state,* similarly to the study by Bąk-Sosnowska et al. [32]. In the study by G. Nowicki and B. Ślusarska, the respondents gave the greatest importance to health as a *property* and as an *objective* [8]. The study by Cybulski et al. also showed that the elderly attached the highest importance to health understood as *property*, then as an *objective, state, process*, and a *result* [14]. Each of the studied groups understood health primary as a property. According to Juczyński, this is an instrumental approach [27]. This fact should not come as a surprise, as the respondents were a sick and hospitalized group, for whom the disease was often an unpleasant experience and a source of stress [8,33,34,35,36].

The analysis of the study results based on the Inventory of Health-Related Behaviour (IHB) showed that amongst subjects the most common health behaviours were related to ‘health practices’, slightly less common in the area of ‘positive mental attitude’ and the least common in the area of ‘proper eating habits’ and ‘preventive behaviours’. Similar observations were made in the studies by Nowicki et al. and Cybulski et al. [8,14]. However, somewhat different results were obtained in studies of elderly people by Prakash et al. and Zanjani et al. There, health behaviours in the area of ‘correct eating habits’ were the most intense [23,24].

When correlating the level of health behaviours according to the IHB with individual categories of the LHC, a total of 59 statistically significant relationships were observed. A positive correlation was found in the group of respondents where the ‘preventive health behaviours’ were more intense; from these, the more important criterion for the respondents was ‘eating properly’. Negative relationships were also found for the respondents for whom ‘correct eating habits’ were important; here, respondents placed more importance on ‘infrequent need of going to the doctor’s’.

The obtained results of the authors’ own study, carried out with the use of the LHC, differed slightly from the results of other cited authors [8,14,15,37,38,39,40,41,42]. According to our respondents aged 50+, ‘being health’ meant above all ‘not feeling any physical ailments’; ‘having all body parts functional’; ‘feeling well’; ‘eating properly’; and ‘infrequent need of going to the doctor’. The differences between the evaluation of health by the elderly presented in the available literature may result from the socio-cultural differences of individual regions of Poland, personal beliefs, general economy, politics, and organization of the society structure [2,8,14,15,35,43]. Cultural determinants—the system of norms and beliefs, patterns of behaviour, as well as all material and non-material products generated by a given social group—play an important role in shaping and choosing healthy behaviours. The determinant of social position is education, which reflects the level of knowledge, including medical knowledge. There was a clear relationship between social status and choice of health behaviours. The higher the education, the higher the requirements for the healthcare, compared to people with lower education [44,45]. Lifestyle plays a fundamental role in maintaining health and prevention of diseases, also in the elderly. Lifestyle elements that have beneficial health effects include correct diet and eating habits, optimal level of physical activity, adequate amount of sleep, satisfactory social relations, skilful use of free time, and knowledge of health prevention [2,8,35,41,46,47,48].

A properly balanced diet is particularly important for maintaining health. Based on the conducted study, the principles of proper nutrition were not perceived by the respondents as a significant factor for preventing or combating already existing cardiovascular diseases. Irrespectively, dietary risk is the most important behavioural health factor in the world, and appears to be the best target in the fight against cardiovascular disease. In the research by Raver A. et al., it has been proven that a proper diet, especially the Mediterranean diet, brings effective results in reducing cardiovascular risk worldwide [49].

However, an incorrect diet and low physical activity are linked to overweight and obesity, which contribute to the occurrence of cardiovascular diseases. Obesity was found in 34.87% of the respondents. Seravalle G. et al. found that obesity, and especially the excessive distribution of visceral fat, was accompanied by several changes at the hormonal, inflammatory, and endothelial levels. These changes induce the stimulation of several other mechanisms that contribute to hypertension and increase cardiovascular morbidity [50].

Given the above, the evidence clearly shows the importance of lifestyle factors (e.g., diet, exercise, and drug use) in the development of cardiovascular disease. Interventions targeted at these behaviours can greatly improve results, and therefore the health and well-being of patients with cardiovascular disease [51].

Summarizing the conducted study, it may be stated that it is important to try and promote health in the hierarchy of values of people aged 50+ and to strengthen their responsibility for their own health and life. Broadly understood education should help raise public health awareness.

As a result of the conducted study, it was found that the most important criteria (according to LHC) that determine health according to the respondents were proper nutrition, adequate sleep, rest, work without stress and tension, and the ability to solve difficult life situations. A particularly high score was given to one of four areas of health behaviours in IHB, namely, ‘health practices’. It proved that the respondents had proper daily health habits, which is very valuable and may determine good health in the future.

The use of standardized questionnaires in further research will allow for a wider observation of health behaviours in people of different ages. Additionally, increasing the number of subjects would help further determine health behaviours, as well as assess the health criteria that the subjects considered as crucial. After more extensive research in the group of people aged 50+ with cardiovascular diseases, more constructive conclusions might be drawn, which would allow specific preventive measures to be taken from the earliest years in order to prevent cardiovascular diseases.

## 5. Conclusions

Respondents aged 50+ and hospitalized for cardiovascular diseases indicated (based on IHB questionnaire) that health behaviours in the area of ‘health practices’ had the strongest correlation with their health, while the lowest correlation was found in the areas of ‘correct eating habits’ and ‘preventive behaviours’. According to the respondents, the most important criteria determining health (according to the LHC Questionnaire) included ’not feeling any physical ailments’; ‘having all body parts functional’; ‘feeling well’; ‘eating properly’; and ‘infrequent need of going to the doctor’.

Based on the information collected from the respondents, it was found that the most important criteria determining health depended on selected descriptive variables, such as age—it was found that for older respondents, avoidance of stimulants is one of the most important factors determining health; gender—the majority of the studied group of women perceived health in many aspects, including feeling happy and wellbeing; place of residence—respondents from urban environment put emphasis on the role of healthy eating in maintaining good health, as opposed to their counterparts from a rural environment; education—respondents with education higher education paid attention to many factors determining health, as opposed to respondents with lower education, e.g., solving life problems and work without stress; and lastly, marital status—married respondents attached great importance to the principles of proper nutrition in order to maintain health compared to the group of respondents who are not married.

Practical implications: Due to the differences in respondents’ preferences in terms of health behaviours, as well as their approach to their health that depended on gender, age, education, and place of residence, it is particularly important to take measures aimed at providing health education to society, which can effectively overcome the problem of anti-health behaviours, existing health imbalance, and help fight cardiovascular disease.

## Figures and Tables

**Table 1 ijerph-18-04221-t001:** Characteristics of the respondents.

Characteristics of the Respondents		N	%
Gender	Women	186	45.53
	Men	223	54.47
	No answer	2	0.49
Age (in years)	N—411	-	-
	Mean—69.2	-	-
	SD—9.45	-	-
	Median—69	-	-
	Min—50	-	-
	Max—93	-	-
	Q1—62	-	-
	Q3—76	-	-
BMI	Underweight [17–18.5]	2	0.49
	Correct weight [18.5–25]	100	24.33
	Overweight [25–30]	157	38.20
	Obesity [30–35]	125	30.41
	Class 2 obesity [35–40]	15	3.65
	Class 3 obesity [>40]	8	1.95
	No data available (weight and/or height)	4	0.97
Place of residence	City < 100,000 inhabitants	223	54.26
	City > 100,000 inhabitants	3	0.73
	Rural area	183	44.53
	No answer	2	0.49
Education	Primary	50	12.17
	Vocational	191	46.47
	Secondary	145	35.28
	Higher	22	5.35
	No answer	3	0.73
Professional activity	Professionally active	84	20.44
	Professionally inactive	313	76.16
	No answer	14	3.41
Marital status	Single	12	2.92
	Married	311	75.67
	In separation	2	0.49
	Divorced	7	1.70
	Widow/widower	73	17.76
	In a partnership	5	1.22
	No answer	1	0.24
Treatment of cardiovascular disease	Hypertension	162	39.42
	Atherosclerosis	240	58.39
	Ischemic heart disease	262	63.75
	Rhythm and cardiac conduction disorders	105	25.55
	Heart defects (congenital and acquired)	29	7.06
	Varicose veins of the lower extremities	109	26.52
	Venous thrombosis	28	6.81
	Other diseases	43	10.46

N—number of respondents; %—percentage of respondents; SD—average age of respondents; Q—quartile.

**Table 2 ijerph-18-04221-t002:** List of Health Criteria of the studied group.

Criterion	N	Mean	SD	Median	Min	Max	Q1	Q3
reaching very old age	411	0.01	0.12	0	0	1	0	0
feeling happy most of the time	411	0.17	0.75	0	0	5	0	0
getting along well with other people	411	0.55	1.09	0	0	5	0	1
knowing how to solve one’s own problems	411	0.45	1.08	0	0	5	0	0
eating properly	411	1.08	1.31	0	0	5	0	2
adequate amount of rest, sleep	411	0.62	0.91	0	0	4	0	1
drinking little or no alcohol	411	0.22	0.75	0	0	5	0	0
not smoking tobacco	411	0.88	1.16	1	0	5	0	1
having a correct body weight	411	0.09	0.48	0	0	4	0	0
taking medication rarely	411	0.27	0.84	0	0	4	0	0
having a good mood	411	0.24	0.85	0	0	5	0	0
not feeling any physical ailments	411	3.11	1.77	4	0	5	2	4
ability to work without tension and stress	411	0.25	0.93	0	0	5	0	0
not getting sick, at most with flu, indigestion (rarely)	411	0.25	0.87	0	0	5	0	0
having healthy eyes, hair and skin	411	0.03	0.29	0	0	4	0	0
ability to adapt to life changes	411	0.48	1.23	0	0	5	0	0
ability to enjoy life	411	0.18	0.77	0	0	5	0	0
being responsible	411	0.05	0.37	0	0	4	0	0
ability to control one’s own feelings and desires	411	0.02	0.19	0	0	3	0	0
having all body parts functional	411	2.45	2.34	3	0	6	0	5
accepting oneself, knowing one’s possibilities and limitations	411	0.4	1.12	0	0	5	0	0
having a job, various interests	411	0.13	0.6	0	0	5	0	0
feeling well	411	1.7	2.01	0	0	6	0	4
infrequent need of going to the doctor	411	1	1.69	0	0	5	0	2

N—number of respondents; SD—average age of respondents; Q—quartile.

**Table 3 ijerph-18-04221-t003:** Correlations of health statements with the age of the respondents.

Criterion	Correlation with Age			
	Correlation Coefficient	*p* *	Correlation Direction	Correlation Strength
reaching very old age	−0.021	*p* = 0.673	---	---
feeling happy most of the time	0.051	*p* = 0.304	---	---
getting along well with other people	0.059	*p* = 0.234	---	---
knowing how to solve one’s own problems	−0.067	*p* = 0.174	---	---
eating properly	0.009	*p* = 0.858	---	---
adequate amount of rest, sleep	−0.032	*p* = 0.523	---	---
drinking little or no alcohol	0.115	*p* = 0.02	positive	very weak
not smoking tobacco	0.026	*p* = 0.593	---	---
having a correct body weight	−0.037	*p* = 0.457	---	---
taking medication rarely	−0.002	*p* = 0.976	---	---
having a good mood	−0.038	*p* = 0.442	---	---
not feeling any physical ailments	0.021	*p* = 0.671	---	---
ability to work without tension and stress	−0.174	*p* < 0.001	negative	very weak
not getting sick, at most with flu, indigestion (rarely)	0.159	*p* = 0.001	positive	very weak
having healthy eyes, hair and skin	0.1	*p* = 0.044	positive	very weak
ability to adapt to life changes	−0.054	*p* = 0.271	---	---
ability to enjoy life	−0.064	*p* = 0.198	---	---
being responsible	−0.055	*p* = 0.268	---	---
ability to control one’s own feelings and desires	−0.068	*p* = 0.166	---	---
having all body parts functional	0.001	*p* = 0.978	---	---
accepting oneself, knowing one’s possibilities and limitations	−0.069	*p* = 0.163	---	---
having a job, various interests	−0.019	*p* = 0.703	---	---
feeling well	0.09	*p* = 0.067	---	---
infrequent need of going to the doctor	0.037	*p* = 0.46	---	---

* *p* = Non-normal distribution of both correlated variables, Pearson correlation coefficient; NP = Non-normal distribution of at least one of the correlated variables, Spearman’s correlation coefficient.

**Table 4 ijerph-18-04221-t004:** Correlations of health statements with the gender of the respondents.

Criterion		Women	Men	*p* *
reaching very old age	Median	0	0	0.806
	Quartile	0–0	0–0	
feeling happy most of the time	Median	0	0	0.048
	Quartile	0–0	0–0	
getting along well with other people	Median	0	0	0.325
	Quartile	0–1	0–0	
knowing how to solve one’s own problems	Median	0	0	0.537
	Quartile	0–0	0–0	
eating properly	Median	0.5	0	0.012
	Quartile	0–3	0–2	
adequate amount of rest, sleep	Median	0	0	0.581
	Quartile	0–1	0–1	
drinking little or no alcohol	Median	0	0	0.002
	Quartile	0–0	0–0	
not smoking tobacco	Median	1	0	0.001
	Quartile	0–1	0–1	
having a correct body weight	Median	0	0	0.76
	Quartile	0–0	0–0	
taking medication rarely	Median	0	0	0.065
	Quartile	0–0	0–0	
having a good mood	Median	0	0	0.995
	Quartile	0–0	0–0	
not feeling any physical ailments	Median	4	4	0.002
	Quartile	0–4	3–5	
ability to work without tension and stress	Median	0	0	0.01
	Quartile	0–0	0–0	
not getting sick, at most with flu, indigestion (rarely)	Median	0	0	0.028
	Quartile	0–0	0–0	
having healthy eyes, hair and skin	Median	0	0	0.857
	Quartile	0–0	0–0	
ability to adapt to life changes	Median	0	0	0.14
	Quartile	0–0	0–0	
ability to enjoy life	Median	0	0	0.261
	Quartile	0–0	0–0	
being responsible	Median	0	0	0.322
	Quartile	0–0	0–0	
ability to control one’s own feelings and desires	Median	0	0	0.855
	Quartile	0–0	0–0	
having all body parts functional	Median	0	4	0.023
	Quartile	0–5	0–5	
accepting oneself, knowing one’s possibilities and limitations	Median	0	0	0.301
	Quartile	0–0	0–0	
having a job, various interests	Median	0	0	0.37
	Quartile	0–0	0–0	
feeling well	Median	0	0	0.283
	Quartile	0–4	0–4	
infrequent need of going to the doctor	Median	0	0	0.001
	Quartile	0–0	0–3	

* *p* = Normal distribution in groups, Student’s *t*-test; NP = Non-normal distribution in groups, Mann–Whitney U test.

**Table 5 ijerph-18-04221-t005:** Correlations of health statements with the BMI of the respondents.

Criterion		Underweight, Correct Weight	Overweight	Obesity	*p* *
reaching very old age	Median	0	0	0	0.249
	Quartile	0–0	0–0	0–0	
feeling happy most of the time	Median	0	0	0	0.203
	Quartile	0–0	0–0	0–0	
getting along well with other people	Median	0	0	0	0.116
	Quartile	0–1	0–1	0–0	
knowing how to solve one’s own problems	Median	0	0	0	0.221
	Quartile	0–0	0–0	0–0	
eating properly	Median	0	1	0	0.025
	Quartile	0–2.75	0–2	0–2	Under.Corr.Over > O
adequate amount of rest, sleep	Median	0	0	0	0.845
	Quartile	0–1	0–1	0–1	
drinking little or no alcohol	Median	0	0	0	0.248
	Quartile	0–0	0–0	0–0	
not smoking tobacco	Median	1	1	0.5	0.544
	Quartile	0–2	0–1	0–1	
having a correct body weight	Median	0	0	0	0.652
	Quartile	0–0	0–0	0–0	
taking medication rarely	Median	0	0	0	0.565
	Quartile	0–0	0–0	0–0	
having a good mood	Median	0	0	0	0.349
	Quartile	0–0	0–0	0–0	
not feeling any physical ailments	Median	4	4	4	0.129
	Quartile	0–4	3–4	2–4	
ability to work without tension and stress	Median	0	0	0	0.236
	Quartile	0–0	0–0	0–0	
not getting sick, at most with flu, indigestion (rarely)	Median	0	0	0	0.769
	Quartile	0–0	0–0	0–0	
having healthy eyes, hair and skin	Median	0	0	0	0.272
	Quartile	0–0	0–0	0–0	
ability to adapt to life changes	Median	0	0	0	0.504
	Quartile	0–0	0–0	0–0	
ability to enjoy life	Median	0	0	0	0.835
	Quartile	0–0	0–0	0–0	
being responsible	Median	0	0	0	0.564
	Quartile	0–0	0–0	0–0	
ability to control one’s own feelings and desires	Median	0	0	0	0.239
	Quartile	0–0	0–0	0–0	
having all body parts functional	Median	0	3	4	0.038
	Quartile	0–5	0–5	0–5	O > Under-Corr
accepting oneself, knowing one’s possibilities and limitations	Median	0	0	0	0.795
	Quartile	0–0	0–0	0–0	
having a job, various interests	Median	0	0	0	0.053
	Quartile	0–0	0–0	0–0	
feeling well	Median	1	0	0	0.235
	Quartile	0–4	0–3	0–4	
infrequent need of going to the doctor	Median	0	0	0	0.01
	Quartile	0–0.75	0–0	0–3	O > Under-Corr.Over

* *p* = Normal distribution in groups, ANOVA + results of post-hoc analysis (Fisher’s LSD test); NP = Non-normal distribution in groups, Kruskal–Wallis test + post-hoc analysis results (Dunn’s test).

**Table 6 ijerph-18-04221-t006:** Correlations of health statements with the place of residence of the respondents.

Criterion		Urban	Rural	*p* *
reaching very old age	Median	0	0	0.796
	Quartile	0–0	0–0	
feeling happy most of the time	Median	0	0	0.316
	Quartile	0–0	0–0	
getting along well with other people	Median	0	0	0.229
	Quartile	0–0	0–1	
knowing how to solve one’s own problems	Median	0	0	0.49
	Quartile	0–0	0–0	
eating properly	Median	1	0	0.003
	Quartile	0–2	0–2	
adequate amount of rest, sleep	Median	0	0	0.011
	Quartile	0–2	0–1	
drinking little or no alcohol	Median	0	0	0.01
	Quartile	0–0	0–0	
not smoking tobacco	Median	1	0	0.551
	Quartile	0–1	0–1	
having a correct body weight	Median	0	0	0.019
	Quartile	0–0	0–0	
taking medication rarely	Median	0	0	0.428
	Quartile	0–0	0–0	
having a good mood	Median	0	0	0.543
	Quartile	0–0	0–0	
not feeling any physical ailments	Median	4	4	0.832
	Quartile	2.25–4	2–4	
ability to work without tension and stress	Median	0	0	0.014
	Quartile	0–0	0–0	
not getting sick, at most with flu, indigestion (rarely)	Median	0	0	0.002
	Quartile	0–0	0–0	
having healthy eyes, hair and skin	Median	0	0	0.223
	Quartile	0–0	0–0	
ability to adapt to life changes	Median	0	0	0.323
	Quartile	0–0	0–0	
ability to enjoy life	Median	0	0	0.591
	Quartile	0–0	0–0	
being responsible	Median	0	0	0.764
	Quartile	0–0	0–0	
ability to control one’s own feelings and desires	Median	0	0	0.071
	Quartile	0–0	0–0	
having all body parts functional	Median	3	3	0.369
	Quartile	0–5	0–5	
accepting oneself, knowing one’s possibilities and limitations	Median	0	0	0.006
	Quartile	0–0	0–0	
having a job, various interests	Median	0	0	0.869
	Quartile	0–0	0–0	
feeling well	Median	0	2	0.01
	Quartile	0–3	0–4	
infrequent need of going to the doctor	Median	0	0	0.001
	Quartile	0–0	0–3	

* *p* = Normal distribution in groups, Student’s *t*-test; NP = Non-normal distribution in groups, Mann–Whitney U test.

**Table 7 ijerph-18-04221-t007:** Correlations of health statements with the education of the respondents.

Criterion		Primary	Vocational	Secondary	Higher	*p* *
reaching very old age	Median	0	0	0	0	0.23
	Quartile	0–0	0–0	0–0	0–0	
feeling happy most of the time	Median	0	0	0	0	0.147
	Quartile	0–0	0–0	0–0	0–0	
getting along well with other people	Median	1	0	0	1	0.001
	Quartile	0–2	0–0	0–1	0–2	H,P > S,V
knowing how to solve one’s own problems	Median	0	0	0	0	0.001
	Quartile	0–1	0–0	0–0	0–2	H > P,V,S
eating properly	Median	0	1	1	0	0.001
	Quartile	0–0	0–2	0–3	0–2	S,V,H > P
adequate amount of rest, sleep	Median	0	0	0	0	0.001
	Quartile	0–0	0–1	0–2	0–0	S,V > H,P
drinking little or no alcohol	Median	0	0	0	0	0.001
	Quartile	0–1	0–0	0–0	0–0	P > S,V,H
not smoking tobacco	Median	0	1	1	0	0.689
	Quartile	0–2	0–1	0–1	0–1.75	
having a correct body weight	Median	0	0	0	0	0.007
	Quartile	0–0	0–0	0–0	0–0	S > V,P,H
taking medication rarely	Median	0	0	0	0	0.334
	Quartile	0–0	0–0	0–0	0–0	
having a good mood	Median	0	0	0	0	0.178
	Quartile	0–0	0–0	0–0	0–0	
not feeling any physical ailments	Median	4.5	4	4	0	0.001
	Quartile	3–5	3–4	2–4	0–2.5	P > V,S,H V,S > H
ability to work without tension and stress	Median	0	0	0	2.5	0.001
	Quartile	0–0	0–0	0–0	0–4	H > S,V,P
not getting sick, at most with flu, indigestion (rarely)	Median	0	0	0	0	0.001
	Quartile	0–0.75	0–0	0–0	0–0	P > V,S,H V > S, H
having healthy eyes, hair and skin	Median	0	0	0	0	0.146
	Quartile	0–0	0–0	0–0	0–0	
ability to adapt to life changes	Median	0	0	0	2	0.001
	Quartile	0–0	0–0	0–0	0–3.75	H >P, S,V
ability to enjoy life	Median	0	0	0	0	0.084
	Quartile	0–0	0–0	0–0	0–0	
being responsible	Median	0	0	0	0	0.747
	Quartile	0–0	0–0	0–0	0–0	
ability to control one’s own feelings and desires	Median	0	0	0	0	0.001
	Quartile	0–0	0–0	0–0	0–0	H > S,P,V
having all body parts functional	Median	0	5	3	0	0.001
	Quartile	0–2	0–5	0–5	0–0	V > S,P,H S > P,H
accepting oneself, knowing one’s possibilities and limitations	Median	0	0	0	0	0.012
	Quartile	0–0	0–0	0–0	0–0	H,S,V > P
having a job, various interests	Median	0	0	0	0	0.047
	Quartile	0–0	0–0	0–0	0–0	H > S,V,P
feeling well	Median	2	0	0	0	0.042
	Quartile	0–4	0–3.5	0–4	0–2.75	P > V,H
infrequent need of going to the doctor	Median	1	0	0	0	0.001
	Quartile	0–2	0–3	0–0	0–0	P > S,H V,S > H

* *p* = Normal distribution in groups, ANOVA + results of post-hoc analysis (Fisher’s LSD test); NP = Non-normal distribution in groups, Kruskal–Wallis test + post-hoc analysis results (Dunn’s test).

**Table 8 ijerph-18-04221-t008:** Correlations of health statements with the professional activity of the respondents.

Criterion		Professionally Active	Professionally Inactive	*p* *
reaching very old age	Median	0	0	0.464
	Quartile	0–0	0–0	
feeling happy most of the time	Median	0	0	0.389
	Quartile	0–0	0–0	
getting along well with other people	Median	0	0	0.041
	Quartile	0–0	0–1	
knowing how to solve one’s own problems	Median	0	0	0.589
	Quartile	0–0	0–0	
eating properly	Median	1	0	0.052
	Quartile	0–2.25	0–2	
adequate amount of rest, sleep	Median	0	0	0.245
	Quartile	0–1.25	0–1	
drinking little or no alcohol	Median	0	0	0.258
	Quartile	0–0	0–0	
not smoking tobacco	Median	1	0	0.566
	Quartile	0–1	0–1	
having a correct body weight	Median	0	0	0.491
	Quartile	0–0	0–0	
taking medication rarely	Median	0	0	0.879
	Quartile	0–0	0–0	
having a good mood	Median	0	0	0.812
	Quartile	0–0	0–0	
not feeling any physical ailments	Median	4	4	0.571
	Quartile	2–4	2–4	
ability to work without tension and stress	Median	0	0	0.012
	Quartile	0–0	0–0	
not getting sick, at most with flu, indigestion (rarely)	Median	0	0	0.001
	Quartile	0–0	0–0	
having healthy eyes, hair and skin	Median	0	0	0.3
	Quartile	0–0	0–0	
ability to adapt to life changes	Median	0	0	0.914
	Quartile	0–0	0–0	
ability to enjoy life	Median	0	0	0.65
	Quartile	0–0	0–0	
being responsible	Median	0	0	0.14
	Quartile	0–0	0–0	
ability to control one’s own feelings and desires	Median	0	0	0.156
	Quartile	0–0	0–0	
having all body parts functional	Median	3.5	3	0.372
	Quartile	0–5	0–5	
accepting oneself, knowing one’s possibilities and limitations	Median	0	0	0.153
	Quartile	0–0	0–0	
having a job, various interests	Median	0	0	0.435
	Quartile	0–0	0–0	
feeling well	Median	0	0	0.105
	Quartile	0–3	0–4	
infrequent need of going to the doctor	Median	0	0	0.109
	Quartile	0–0	0–2	

* *p* = Normal distribution in groups, Student’s *t*-test; NP = Non-normal distribution in groups, Mann–Whitney U test.

**Table 9 ijerph-18-04221-t009:** Correlations of health statements with the marital status of the respondents.

Criterion		Married	Other	*p* *
reaching very old age	Median	0	0	0.668
	Quartile	0–0	0–0	
feeling happy most of the time	Median	0	0	0.024
	Quartile	0–0	0–0	
getting along well with other people	Median	0	0	0.007
	Quartile	0–0	0–1	
knowing how to solve one’s own problems	Median	0	0	0.776
	Quartile	0–0	0–0	
eating properly	Median	1	0	0.001
	Quartile	0–2	0–1	
adequate amount of rest, sleep	Median	0	0	0.002
	Quartile	0–1	0–0	
drinking little or no alcohol	Median	0	0	0.006
	Quartile	0–0	0–0	
not smoking tobacco	Median	1	0	0.027
	Quartile	0–1	0–1	
having a correct body weight	Median	0	0	0.326
	Quartile	0–0	0–0	
taking medication rarely	Median	0	0	0.856
	Quartile	0–0	0–0	
having a good mood	Median	0	0	0.787
	Quartile	0–0	0–0	
not feeling any physical ailments	Median	4	3	0.036
	Quartile	2.5–4	0–4	
ability to work without tension and stress	Median	0	0	0.036
	Quartile	0–0	0–0	
not getting sick, at most with flu, indigestion (rarely)	Median	0	0	0.036
	Quartile	0–0	0–0	
having healthy eyes, hair and skin	Median	0	0	0.226
	Quartile	0–0	0–0	
ability to adapt to life changes	Median	0	0	0.167
	Quartile	0–0	0–0	
ability to enjoy life	Median	0	0	0.611
	Quartile	0–0	0–0	
being responsible	Median	0	0	0.769
	Quartile	0–0	0–0	
ability to control one’s own feelings and desires	Median	0	0	0.259
	Quartile	0–0	0–0	
having all body parts functional	Median	4	0	0.002
	Quartile	0–5	0–4	
accepting oneself, knowing one’s possibilities and limitations	Median	0	0	0.044
	Quartile	0–0	0–0	
having a job, various interests	Median	0	0	0.326
	Quartile	0–0	0–0	
feeling well	Median	0	2	0.001
	Quartile	0–3	0–4	
infrequent need of going to the doctor	Median	0	0	0.25
	Quartile	0–1	0–2	

* *p* = Normal distribution in groups, Student’s *t*-test; NP = Non-normal distribution in groups, Mann–Whitney U test.

**Table 10 ijerph-18-04221-t010:** Degree of intensity of specific categories of health behaviours according to the Inventory of Health-Related Behaviour (IHB) amongst respondents.

IHB Sub-Scales	N	Mean	SD	Median	Min	Max	Q1	Q3
Correct eating habits	411	3.28	0.75	3.17	1	5	2.83	4
Preventive behaviours	411	3	0.62	3	1.17	4.5	2.67	3.67
Positive mental attitude	411	3.59	0.52	3.83	1.33	4.67	3.33	4
Health practices	411	3.61	0.53	3.67	1.67	4.83	3.33	4

**Table 11 ijerph-18-04221-t011:** The level of health behaviours according to the Inventory of Health-Related Behaviour (IHB) versus number of selections of individual categories of the List of Health Criteria (LHC).

LHC	IHB	Correlation Coefficient	*p*	Correlation Direction	Correlation Strength
Eating properly	Preventive behaviours	0.521	*p* < 0.001	positive	Average
Eating properly	Overall IHB result	0.492	*p* < 0.001	positive	Weak
Eating properly	Correct eating habits	0.414	*p* < 0.001	positive	Weak
Eating properly	Positive mental attitude	0.414	*p* < 0.001	positive	Weak
Adequate amount of rest, sleep	Preventive behaviours	0.407	*p* < 0.001	positive	Weak
Infrequent need of going to the doctor	Correct eating habits	−0.392	*p* < 0.001	negative	Weak
Infrequent need of going to the doctor	Overall IHB result	−0.354	*p* < 0.001	negative	Weak
Adequate amount of rest, sleep	Overall IHB result	0.332	*p* < 0.001	positive	Weak
Not smoking tobacco	Positive mental attitude	0.315	*p* < 0.001	positive	Weak
Adequate amount of rest, sleep	Positive mental attitude	0.313	*p* < 0.001	positive	Weak
Not smoking tobacco	Overall IHB result	0.302	*p* < 0.001	positive	Weak
Eating properly	Health practices	0.296	*p* < 0.001	positive	very weak
Feeling well	Preventive behaviours	−0.295	*p* < 0.001	negative	very weak
Feeling well	Overall IHB result	−0.291	*p* < 0.001	negative	very weak
Infrequent need of going to the doctor	Preventive behaviours	−0.285	*p* < 0.001	negative	very weak
Not getting sick, at most with flu, indigestion (rarely)	Overall IHB result	−0.279	*p* < 0.001	negative	very weak
Feeling well	Positive mental attitude	−0.276	*p* < 0.001	negative	very weak
Infrequent need of going to the doctor	Health practices	−0.274	*p* < 0.001	negative	very weak
Not smoking tobacco	Preventive behaviours	0.266	*p* < 0.001	positive	very weak
Not getting sick, at most with flu, indigestion (rarely)	Preventive behaviours	−0.263	*p* < 0.001	negative	very weak
Having all body parts functional	Preventive behaviours	0.263	*p* < 0.001	positive	very weak
Not getting sick, at most with flu, indigestion (rarely)	Correct eating habits	−0.262	*p* < 0.001	negative	very weak
Feeling well	Correct eating habits	−0.262	*p* < 0.001	negative	very weak
Adequate amount of rest, sleep	Correct eating habits	0.251	*p* < 0.001	positive	very weak
Not getting sick, at most with flu, indigestion (rarely)	Positive mental attitude	−0.239	*p* < 0.001	negative	very weak
Infrequent need of going to the doctor	Positive mental attitude	−0.238	*p* < 0.001	negative	very weak
Knowing how to solve one’s own problems	Positive mental attitude	−0.232	*p* < 0.001	negative	very weak
Not smoking tobacco	Health practices	0.229	*p* < 0.001	positive	very weak
Getting along well with other people	Preventive behaviours	−0.224	*p* < 0.001	negative	very weak
Having all body parts functional	Positive mental attitude	0.22	*p* < 0.001	positive	very weak
Not smoking tobacco	Correct eating habits	0.217	*p* < 0.001	positive	very weak
Ability to adapt to life changes	Positive mental attitude	−0.196	*p* < 0.001	negative	very weak
Having all body parts functional	Overall IHB result	0.186	*p* < 0.001	positive	very weak
Knowing how to solve one’s own problems	Preventive behaviours	−0.181	*p* < 0.001	negative	very weak
Not getting sick, at most with flu, indigestion (rarely)	Health practices	−0.167	*p* = 0.001	negative	very weak
Taking medication rarely	Correct eating habits	−0.158	*p* = 0.001	negative	very weak
Taking medication rarely	Health practices	−0.157	*p* = 0.001	negative	very weak
Getting along well with other people	Positive mental attitude	−0.148	*p* = 0.003	negative	very weak
Ability to work without tension and stress	Correct eating habits	0.142	*p* = 0.004	positive	very weak
Ability to adapt to life changes	Preventive behaviours	−0.139	*p* = 0.005	negative	very weak
Taking medication rarely	Overall IHB result	−0.136	*p* = 0.006	negative	very weak
Feeling well	Health practices	−0.133	*p* = 0.007	negative	very weak
Ability to enjoy life	Positive mental attitude	0.13	*p* = 0.008	positive	very weak
Being responsible	Overall IHB result	−0.129	*p* = 0.009	negative	very weak
Taking medication rarely	Preventive behaviours	−0.123	*p* = 0.013	negative	very weak
Being responsible	Preventive behaviours	−0.123	*p* = 0.013	negative	very weak
Knowing how to solve one’s own problems	Overall IHB result	−0.122	*p* = 0.013	negative	very weak
Adequate amount of rest, sleep	Health practices	0.122	*p* = 0.013	positive	very weak
Ability to control one’s own feelings and desires	Correct eating habits	0.121	*p* = 0.014	positive	very weak
Being responsible	Positive mental attitude	−0.114	*p* = 0.021	negative	very weak
Being responsible	Correct eating habits	−0.113	*p* = 0.021	negative	very weak
Ability to enjoy life	Overall IHB result	0.109	*p* = 0.027	positive	very weak
Getting along well with other people	Overall IHB result	−0.104	*p* = 0.035	negative	very weak
Reaching very old age	Health practices	−0.103	*p* = 0.036	negative	very weak
Feeling happy most of the time	Correct eating habits	0.102	*p* = 0.039	positive	very weak
Ability to work without tension and stress	Preventive behaviours	0.102	*p* = 0.04	positive	very weak
Ability to enjoy life	Correct eating habits	0.1	*p* = 0.044	positive	very weak
Ability to enjoy life	Health practices	0.099	*p* = 0.045	positive	very weak
Reaching very old age	Correct eating habits	−0.098	*p* = 0.046	negative	very weak

* *p* = Normal distribution of both correlated variables, Pearson correlation coefficient; NP = Non-normal distribution of at least one of the correlated variables, Spearman’s correlation coefficient.

## Data Availability

Data are not publicly available and data sharing is not applicable to this article.

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
