# Peer review of "Assessment of the Health Behaviours and Value-Based Health Analysis of People Aged 50+ Who Were Hospitalized Due to Cardiovascular Disease"

_ijerph, 2021, doi:10.3390/ijerph18084221_

Round 1

Reviewer 1 Report

This study aims to measure health behaviours and value-based health analysis made by people aged 50+ who have been hospitalized due to cardiovascular disease. The paper has major limitation which are outlined below and which, in my opinion, impede its publication.

Abstract

Line 15: "Human health depends mainly on individual health behaviours". The tone of this sentence should be toned down. Strictly speaking, the meaning of the sentence is not correct, as there are many other factors, not only behavioural, that influence health status.

Line 16: do not repeat "depends" in two consecutive sentences.

Line 17: the objective should be in the infinitive. For example: "To assess...".

Line 18: a space is missing after "disease".

Keywords: to achieve greater visibility of the article, the keywords should not be contained in the title of the article.

Introduction

The introduction needs to be fully revised and restructured by the authors. The introduction does not adequately review the existing literature, nor does it adequately justify the need for this study.

Lines 39-41: the introduction cannot begin with this paragraph. Restructure the introduction for better understanding. It should begin with a background on the state of the subject, and then justify the problem of the study.

Lines 44-46: articles should not be referenced in the way the authors do. The referencing rules established by the journal must be followed.

Lines 47-49: It should be mentioned that health status, although conditioned by behavioural factors, is also influenced by other factors (genetic, for example).

Materials and methods

Authors are advised to subdivide this section for better understanding.

Line 103: include LHC reference.

Lines 139-140: reference the article according to journal guidelines (delete title, authors...).

Results

The results are too long and difficult to understand.

Line 146: there is an error with the percentages of men-women.

Line 195: bullets should not be used in the document.

Discussion

The discussion should begin by summarizing the most important findings of the study.

In general, the discussion repeats the results and the introduction. The authors should focus on discussing their results with those found in the literature, explaining the reasons why these results have been found.

In my opinion, the discussion needs to be substantially improved.

Reviewer 2 Report

Dear Authors,

firstly I would like to thank you for submitting your paper. The topic is quite interesting and always current.

Analyzing the perception of health and healthy behaviors of patients can represent an important tool to better direct the efforts of health professionals in the prevention strategies of cardiovascular diseases that must be increasingly individualized.

The manuscript presents, besides a good starting idea, an in-depth statistical analysis, a correct description of the results and a precise presentation of the conclusions deriving from the study.

However, I have some suggestions:

  • the introduction is too repetitive and redundant and does not specify the reason for the choice of the LHC and IHB questionnaires compared to other analogues available in literature
  • the section methods does not make clear the selection of the patients to whom the questionnaires were administered: were the patients admitted to the reference center consecutively enrolled?
    Furthermore, the pathologies affecting the enrolled patients are not specified: the answers to the questionnaires could be influenced by the different severity of the cardiovascular pathology
  • the discussion section is not easy to read and does not present an explanatory and in-depth evaluation of the results obtained
  • an identification of the limitations of the paper is completely absent (e.g. the study is conducted on a population of hospitalized patients and the results may not be extended to the general population; possible selection bias, etc.)
  • there are some typos that must be corrected and references 1, 8 and 9 must be reviewed.

Reviewer 3 Report

Comments:

  1. Abstract section: Not all  conclusions are supported by results, which are presented in the abstract.  
  2. The description of table 1 is s too long. It should be shortened e.g. "The Characteristics of the respondents is presented in table 1"
  3. Line 177, 191 “SD – average age of respondents”? In my opinion standard deviation may be abbreviated SD.
  4. Please provide information on how the answers (LHC and IHB) were scored in the questionnaires e.g. a score from 0 to 6 (what does mean e.g. zero in the tables)?
  5. How did you calculate the p values (Non-normal distribution in groups, Mann-Whitney test) when median and quartiles are zero e.g. table 4 (drinking little or no alcohol)?
  6. In order to shorten the results I propose to include significant results only in tables from 3 to 9 (all answers are included in table 2). Too many results were shown and described in the results section, which makes the manuscript difficult to read. The authors should show the most important results, (these results, which are discussed). 
  7. All results have non-normal distribution in groups (NP) in tables from 3 to 9. So this information should be include under the tables (Remove NP from tables).
  8. Tables from 4 to 9 don’t show the correlations results (correlation coefficient). It would be better to change the word correlation to e.g. “association between health statements and the gender of the respondents”.  
  9. It would be good to show the more important findings of this study by the graph.

Reviewer 4 Report

Dear authors, 

This is an interesting and relevant study. However, this paper needs several changes to facilitate the understanding of the study and increase the methodological quality.

Comments and suggestions for authors

Title

  1. The title and the objective are practically equal. Moreover, the title is too long, please making it more specific.

Abstract

  1. Line 16: The objective should start with an infinitive verb. Considering that your study is a cross sectional study, your objective may start with “To describe”- Please, rephrase the objective.
  2. Line 18-22: Methods section is one of the most important sections of your work. It should start with the description of your study design. In addition, a more detailed description of the study population (where are these participants from? Hospitals? Health centres? Exclusion criteria?), questionnaires (cut off scores?), statistical analyses (descriptive analyses? Regression models?) and, if it is possible, ethical aspects (informed consent?) is necessary.
  3. Line 22-30: Results are difficult to read. I suggest you to include p-values in this section.
  4. Line 30-34: Conclusions should be “an answer” to the objective. Please, provide a specific conclusion which include health behaviours and value-based health analysis aspects.

Introduction

  1. Line 39-45: These phrases are isolated. I suggest you use them as justification in the last paragraph of your introduction.
  2. Lines 77-84: Please, include some references.
  3. Line 90: The objective should start with an infinitive verb. Considering that your study is a cross sectional study, your objective may start with “To describe”- Please, rephrase the objective.
  4. It may be helpful for readers to include your hypothesis or research question.

Methods

  1. Line 94: Please, start with the study design.
  2. Make subheadings such as “study design”, “study population” (participants selection and recruitment? Cardiovascular diseases description?), “study variables” (which are your main study variables? and include a description on questionnaires scores and their interpretation; which are your other variables and how were them categorized? For example, how was BMI collected and why did you use these categories?) and “statistical analyses” could facilitate the reader’s understanding and this section organization.

Results

There is too much information in this section. This fact makes the results very difficult to understand and read. In my opinion, you should specify your objective and reduce the number of tables because it is not understood whether your purpose of study is to describe the determinants associated with health behaviours or to validate any of the questionnaires you used.

  1. Line 145: “ The overview of the study group”. Do not perform a list of your results. Try to rewrite this paragraph avoiding the use of : , performing a linked and related writing.
  2. Table 1: Remove all % signs from your results as it already appears in the header.
  3. Line 179: Is it possible that subheading “B” is missing here? If not, where is the next section of results?
  4. Tables 4,5,6,7,8,9. Please, change mean+SD for mean(SD), this is the correct description result.
  5. Line 324, other tables footnotes and methods section: The correct name for “Mann-Whitney test” is “Mann-Whitney U test”. Please, correct this throughout the text.

Discussion

  1. Lines 366, 374, 375, 382…: once you have presented an abbreviation for a term, you have to use it throughout the text.
  2. Please, include a study limitations section.

Conclusion

  1. As in the abstract, the conclusion section is not entirely adequate. This section should include a "direct response" to the objective. However, the terms health behaviours and value-based health analysis do not appear once in these conclusions. Please, rewrite the conclusion.

References

  1. Please revise the format of the references 7,11,26, 44 and 46.

Round 2

Reviewer 1 Report

In my opinion, the article remains difficult to follow. The corrections I made in the first round of review have mostly not been resolved. The authors claim that many of the corrections I made to them should not be remedied since the other reviewers have not pointed them out to them, which to me is a big mistake. 
Therefore, I stand by my decision to reject the article. 
Thank you very much for giving me the opportunity to review the article again.
Best regards. 

Reviewer 2 Report

Dear Authors,

thank you for submitting your edited version of the study.

The changes made to the paper in the introduction, results and discussion sections and the refinement of the description of the methods have increased the quality and readability of the manuscript.

However, some typos are still present (for example "to access" instead of "to assess" in line 19 of the abstract) and some references need to be revised (references number 1, 8, 9 and 46) and the limitations section would require a more in-depth analysis.

Author Response

Review no. 2

The authors would like to thank for the review, valuable tips and comments. The authors have implemented all of the Reviewers’ suggestions.

Below we present explanation regarding the Reviewer’s comments:

  • The authors have reviewed the article for typing errors, for example ‘to access’ instead of ‘to assess’ in line 19.
  • The authors reviewed the literature in terms of the proper convention (1, 8, 9, and 46).
  1. Kino, S.; Bernabé, E.; Sabbah, W. Socioeconomic inequality in clusters of health-related behaviours in Europe: latent class analysis of a cross-sectional European survey.BMC Public Health. 2017,17,497-451, doi: 10.1186/s12889-017-4440-3.
  2. Nowicki, G.J.; Ślusarska, B.; Piasecka, H.; Bartoszek, A.; Kocka, K.; Deluga, A. The Status of Cardiovascular Health in Rural and Urban Areas of Janów Lubelski District in Eastern Poland: A Population-Based Study. Int J Environ Res Public Health. 2018,15,2388-2392, doi: 10.3390/ijerph15112388.
  3. Cabellos-García, A.C.; Castro-Sánchez, E.; Martínez-Sabater, A.; Díaz-Herrera, M.Á.; Ocaña-Ortiz, A.; Juárez-Vela, R.; Gea-Caballero, V. Relationshipbetween Determinants of Health, Equity, and Dimensions of Health Literacy in Patients with Cardiovascular Disease. Int J Environ Res Public Health. 2020, 17,2082-2085, doi: 10.3390/ijerph17062082.
  4. Hon, L.T.; Eliza, M.L.; Wong, C.Effectiveness of Educational Interventions on Adherence to Lifestyle Modifications Among Hypertensive Patients: An Integrative Review. Int J Environ Res Public Health 2020,17,2513-2519, doi: 10.3390/ijerph17072513.

  • The Limitations section has been revised (incl. suggestions of another Reviewer for changes in Discussion and Limitations).

The authors would like to inform that the remaining changes introduced in the article are a result of the remaining 3 assessments made by other Reviewers, which the authors fully complied with.

The authors would like to humbly ask for acceptance of this paper.

Reviewer 3 Report

Thank you very much for responding to  my comments. Unfortunately, I still see some flaws.

  1. Abstract: Your conclusion duplicate the results. All results should be shortly summarized in the conclusion.
  2. Line 52 to 55 (based on…). This conclusion isn’t supported by the results described in the abstract section.
  3. SD – average age of respondents (footnotes of table 1 or 2)? Is it standard deviation??
  4. In statistics, the standard deviation (SD) is a measure of the amount of variation or dispersion of a set of values. A low standard deviation indicates that the values tend to be close to the mean of the set, while a high standard deviation indicates that the values are spread out over a wider range. Why did you use SD to comparison your results (abstract section or discussion section)?
  5. Your p-values were calculated by parametric or non-parametric tests? This should be clearly indicated in the tables (if you didn’t use parametric tests please remove footnote “P”). Mean ±SD (parametric tests) and median with quartiles (non-parametric tests).
  6. This information could be included in the description of research tools (line 180), e.g ‘0’ means zero points, i.e. a situation in which a given criteria was not selected by the respondent.  This may facilitate the interpretation of your results. (full scoring would be more favorable).
  7. Why did you remove the word “mean” from the tables?

Reviewer 4 Report

Dear authors,

Thank you for sending us a new version of your work. It is a pertinent and relevant topic of study. However, this manuscript still needs several changes.  

  1. Abstract: The objective is still wrong. I suggest the next change: To assess health behaviors and analyze health, it was assumed that the main objective of the study is the assessment of health behaviors and value-based health analysis made by people aged 50+ who have been hospitalized due to cardiovascular disease. of various levels of severity, based on standardized research tools.
  2. Abstract: Specify the statistical analysis used.
  3. Abstract: SD is not very helpful in this context, you should indicate rho or p-value.
  4. Introduction: The introduction has to be restructured. The first paragraph must be a description on the state of the subject. The introduction is difficult to read.
  5. Methods: Study design is not a part of “research tools”. The study design usually appears in the first paragraph of the methodology.
  6. Methods: What are the scores or puntuaction of this scales? This is very important to interpret the results and tables.
  7. Methods: line 228: Put this reference like the rest.
  8. Methods: You said that you used 2 evaluation tools, however in methods you indicate: lines 188: The third tool was the Inventory of Health-Related Behaviours (IHB) by Z. Juczyński [27]
  9. Results: The results need to start with a short paragraph about describing the results from Table 1.
  10. Results: Are hard to read and understand. In order to shorten the results you could only describe significant results from tables 3-9.
  11. Results: Try to avoid the use of bullet points (line 307)
  12. Results: To show data in tables like this 0 .01±0 .1 is an error, you should show them as mean(SD) à01(0.1). However, in this case as the variables presented have a non-parametrical distribution, you must only present median(IQR).
  13. Results: I do not quite understand the usefulness of tables 4,5,6,7,8,9. Evaluating the correlation between different variables and one of the scales that you used is not part of your objective.
  14. Discussion: This section should starts with a summary of your most important findings.
  15. Discussion: In this section you have to include a limitations paragraph in which you describe methodological aspects and risk of biais that your work can present. This is not enough:

Limitations: Future studies should evaluate the health behaviors of patients with 973 other chronic diseases, and make appropriate comparisons in order to determine health 974 behaviors which pose the greatest threats to the health of the people aged 50+.

In fact, these are the ”future directions” of these research and could be the last sentence of your conclusion.

  1. References: Please revise the format of the references 7,9,16,27,32,38 and 44, they dont have the same year, pages and volume format than the rest of them.
